# Indoloquinoline-Mediated Targeted Downregulation of KRAS through Selective Stabilization of the Mid-Promoter G-Quadruplex Structure

**DOI:** 10.3390/genes13081440

**Published:** 2022-08-13

**Authors:** Alexandra Maria Psaras, Rhianna K. Carty, Jared T. Miller, L. Nathan Tumey, Tracy A. Brooks

**Affiliations:** 1Pharmaceutical Sciences, School of Pharmacy and Pharmaceutical Sciences, Binghamton University, Binghamton, NY 13902, USA; 2BioMolecular Sciences, School of Pharmacy, University of Mississippi, University, MS 38677, USA

**Keywords:** KRAS, G-quadruplex, indoloquinolines, pancreatic cancer

## Abstract

KRAS is a well-validated anti-cancer therapeutic target, whose transcriptional downregulation has been demonstrated to be lethal to tumor cells with aberrant KRAS signaling. G-quadruplexes (G4s) are non-canonical nucleic acid structures that mediate central dogmatic events, such as DNA repair, telomere elongation, transcription and splicing events. G4s are attractive drug targets, as they are more globular than B-DNA, enabling more selective gene interactions. Moreover, their genomic prevalence is increased in oncogenic promoters, their formation is increased in human cancers, and they can be modulated with small molecules or targeted nucleic acids. The putative formation of multiple G4s has been described in the literature, but compounds with selectivity among these structures have not yet been able to distinguish between the biological contribution of the predominant structures. Using cell free screening techniques, synthesis of novel indoloquinoline compounds and cellular models of KRAS-dependent cancer cells, we describe compounds that choose between KRAS promoter G4_near_ and G4_mid_, correlate compound cytotoxic activity with KRAS regulation, and highlight G4_mid_ as the lead molecular non-canonical structure for further targeting efforts.

## 1. Introduction

KRAS is a 21-kD GTPase that plays a role in cell survival, proliferation and differentiation [1,2]. It is constitutively expressed, but active only when GTP-bound. Normal functioning KRAS has a relatively short, and inducible, GTP-bound life. Mutations in RAS proteins are found in approximately one-third of all human tumors, with KRAS being the most frequently mutated isoform [3,4]. Single point mutations of the KRAS gene abolish inherent GTP hydrolysis, rendering the protein constitutively active. The highest incidence of mutational activation occurs in lung, colorectal and pancreatic cancers [1,4,5], where KRAS mutations are associated with increased tumorigenicity and poor prognosis [3]. Mutation of the *KRAS* gene has been identified as a transforming oncogenic event, creating an unstable environment that allows for further mutational selective and increasingly aggressive disease [6]. In the absence of a mutation, increased KRAS activity occurs via gene amplification, overexpression or increased upstream activation [3]. The genomic activation of KRAS, in particular, is associated with metastatic disease and poor prognosis in hormone-related cancers, such as breast, endometrial and ovarian cancer [7,8,9,10,11,12,13].

KRAS is a well validated anti-cancer therapeutic target. There are many ongoing drug discovery programs with a primary focus on individual mutant KRAS isoforms. Sotorasib is a KRAS G12C-targeting drug that was FDA approved in May of 2021 for the treatment of lung cancers that harbor mutant KRAS [14,15]. G12C is not a common mutation in pancreatic or colorectal cancers [16,17], and sotorasib does not benefit patients with amplified or overexpressed KRAS. So, while the development and approval of this drug is remarkable, there remains an unmet clinical need. With efforts to develop compounds for each KRAS mutation, cancers that harbor aberrant KRAS signaling in the absence of a protein mutation have still not been addressed. Notably, transcriptional downregulation of *KRAS* has been demonstrated to be lethal to tumor cells with aberrant KRAS signaling, irrespective of mutational status, and to have a wide potential therapeutic window [1,18,19,20,21]. The formation of non-canonical G-quadruplex (G4) DNA structures within KRAS’s core promoter offers a molecular target to decrease gene expression, utilizing a small molecule with promise in cancer cells that harbor aberrant KRAS signaling and in a manner independent of the mutational status.

A single-strand of guanine-rich DNA can fold upon itself to form a higher order, non-canonical, four-stranded G4. This globular structure arises when four guanines form Hoogsteen hydrogen bonds in a planar arrangement, which then self-stack to form a higher order structure, either inter- or intra-molecularly [22]. Within DNA, intramolecular G4 formation occurs in telomeres, promoter regions, at origins of replication, regions of DNA breaks and locations of DNA repair, within exons to direct splicing, and more [23,24,25,26,27]. G4 formation within promoter regions requires the opening of double-stranded (ds) DNA, which occurs under negative torsional stress induced by transcription [28]. Thus, genes with higher transcription rates have greater potential for promoter G4 formation, underscoring the apparent role of most promoter G4s as negative regulators of gene expression. G4 formation is inducible by small molecules, endogenous levels of G4s can predict tumor sensitivity to G4-targeted ligands, the structures occur more frequently in human tumors, and were recently described to correlate with oncogenesis of B-cells [29,30].

The core promoter of *KRAS* is highly G/C-rich (up to 75%), putatively capable of forming multiple higher order non-B-DNA structures [31,32,33]. The near G4-forming region (G4_near_) has been examined and described by multiple groups to form a parallel structure with three stacked tetrads connected by loops, varying from one to twelve nucleotides and incorporating an interrupted guanine run with a thymine kink. The mid G4-forming region (G4_mid_) has been reported to form a mixed parallel/antiparallel structure, formed by three stacked tetrads connected with loops of two to ten nucleotides. The far G4-forming region does not appear to form a biologically relevant intramolecular G4 [34,35,36,37,38]. The literature has not agreed on the biological activity of each of these structures, relying either on biochemical manipulation of the sequences to abrogate G4 formation or on non-selective compounds. The former approach has the inherent obstacle of DNA mutations that also interrupt transcription factor binding, making interpretation of results more difficult. The latter approach makes it difficult to ascertain the biological role of each G4 to *KRAS* transcription. In the current work, we describe compounds that choose between the *KRAS* promoter G4_near_ and G4_mid_ structures, using the *MYC* promoter G4 as a comparative known entity, discuss the synthesis and activity of novel indoloquinolone compounds, and highlight the potential of these compounds in combination with chemotherapy in pancreatic cancer cell lines.

## 2. Materials and Methods

### 2.1. Chemicals, Compounds, and Oligonucleotides

All oligonucleotides were synthesized and purchased from Eurofins MWG Operon, LLC (Louisville, KY, USA) (Table 1). The NCI Diversity Set III was obtained from the NCI Developmental Therapeutics Program (NCI DTP, Bethesda, MD, USA). Acrylamide/bisacrylamide (29:1) solution and ammonium persulfate were purchased from Bio-Rad laboratories (Hercules, CA, USA), and N,N,N′,N′-tetramethylethylenediamine was purchased through Fisher Scientific (Pittsburgh, PA, USA). All other chemicals were purchased from Sigma-Aldrich (St. Louis, MO, USA).

### 2.2. Chemical Synthesis

**(2-Methoxy-5-nitrophenyl)methanol (2)**: 2-(hydroxymethyl)-4-nitrophenol (500 mg, 2.96 mmol) was added to DMF (10 m) and K_2_CO_3_ (1.23 g, 8.89 mmol). After stirring for 15 min at rt, methyl iodide (276 μL, 4.43 mmol) was added and the mixture was heated to 80 °C. After 19 h, the reaction was removed from heat and the reaction was partitioned between EtOAc and water. The organic layer was dried over MgSO_4_ and concentrated to give (2-methoxy-5-nitrophenyl)methanol (452 mg, 84%). HPLC rt = 1.42 min; *m/z* 166.1 [M-OH]^+^.

**(5-Amino-2-methoxyphenyl)methanol (3)**: (2-methoxy-5-nitrophenyl)methanol (0.81 g, 4.45 mmol) was dissolved in 37 mL of EtOH. Ammonium chloride (2.4 g, 44 mmol, 10 eq) was added to the reaction as a solution in 9 mL of water. After stirring briefly, Fe (2.47 g, 44 mmol, 10 eq) was added and the reaction was heated to reflux overnight. The crude mixture was filtered through celite and the product was extracted into EtOAc. The organic layer was dried over MgSO_4_ and evaporated, giving 668 mg of the title product (98%). HPLC rt = 0.20 min; *m/z* 154.2 [M+H]^+^.

**7-Chloro-3-iodoquinolin-4-ol (5)**: 7-chloroquinolin-4-ol (1.0 g, 5.59 mmol) was suspended in acetic acid (40 mL) and treated with 1-iodopyrrolidine-2,5-dione (1.4 g, 6.1 mmol). The reaction was heated to 60 °C for 45 min, at which time LCMS analysis indicated that the reaction had proceeded to completion. Water (40 mL) was added and the reaction was cooled, allowing the product to crystalize. The title compound was filtered in vacuo and placed in a vacuum desiccator overnight, resulting in 2.1 g of crude material that was used without further purification. HPLC rt = 1.75 min; *m/z* 305.8 [M+H]^+^.

**4,7-Dichloro-3-iodoquinoline (6)**: The crude material from above (~5.6 mmol) was treated directly with POCl_3_ (30.5 mL, 100 eq) and DMF (50 μL, 0.6 mmol). The reaction was heated to 80 °C overnight and then carefully quenched over ice. (Note: the reaction and quenching were performed behind a blast shield). The resulting solid was filtered and dried under vacuum to give the title compound (754 mg, 71% over two steps). HPLC rt = 3.61 min; *m/z* 323.9 [M+H]^+^.

**(3-((7-Chloro-3-iodoquinolin-4-yl)amino)phenyl)methanol (7)**: The 4,7-dichloro-3-iodoquinoline (**6**, 53 mg, 0.16 mmol) was added to a high-pressure vessel and treated with 2 mL of EtOH, m-aminobenzylalcohol (**3**, 28.2 mg, 1.42 eq.) and pyridine hydrochloride (22.0 mg, 1.18 eq.) The vessel was sealed and heated to 90 °C for 2.5 h. An additional 0.45 eq. each of **3** and pyridine hydrochloride was added and the heating continued for 45 min. Upon cooling, the reaction mixture was diluted with water and the product was extracted into EtOAc (3 × 20 mL). The extracts were dried over MgSO_4_ and concentrated to dryness, giving 51 mg (77%) of the title compound. HPLC rt = 2.48 min; *m/z* 411.1 [M+H]^+^.

**(3-Chloro-8-methoxy-11H-indolo [3,2-c]quinolin-9-yl)methanol (8)**: (5-((7-chloro-3-iodoquinolin-4-yl)amino)-2-methoxyphenyl)methanol (**7**, 201.8 mg, 1 Eq, 457.9 µmol) was dissolved into THF (5.00 mL) and water (500 µL). The solution was stirred at room temperature (RT) and treated with cesium carbonate (450.6 mg, 3.02 Eq, 1.38 mmol), triethylamine (95 mg, 0.13 mL, 2.0 Eq, 0.94 mmol) and PdCl_2_ (dppf) (39.3 mg, 0.10 Eq, 48 µmol). The reaction was heated to 90 °C overnight and then stirred, until partitioning between EtOAc (50 mL) and water was carried out (50 mL). The aqueous layer was extracted three times with EtOAc, and three additional times with DCM/MeOH. The combined extracts were dried over MgSO_4_ and concentrated to dryness, giving 113 mg of crude product (~79%). The mixture was purified by preparative HPLC, using a C18 column with a 15–50% ACN/water gradient. The first eluting isomer was determined by NMR to be structure **8**. Typical isolated yield of the single isomer product **8** was ~40%. HPLC rt = 1.33 min; *m/z* 313.1 [M+H]^+^.

Step 1: (3-chloro-8-methoxy-11H-indolo [3,2-c]quinolin-9-yl)methanol (**8**, 10.6 mg, 1 Eq, 33.9 µmol) was dissolved in DMF (0.95 mL) and subsequently treated with pyridinium dichromate (25.5 mg, 2.0 Eq). After stirring for 2 days, the mixture was filtered through a sintered glass funnel and used in the next step without further purification. HPLC rt = 2.54 min; *m/z* 311.1 [M+H]^+^.

Step 2: The aldehyde from step 1 (in 0.5 mL DMF) was treated with sodium cyanoborohydride (13.9 mg, 6.55 Eq, 221 µmol) and diethylamine (15 mg, 22.0 µL, 6.2 Eq, 0.21 mmol), followed by a catalytic amount of AcOH (1.1 mg, 1.0 µL, 0.52 Eq, 17 µmol). The reaction was heated to 80 °C overnight. The mixture was purified by preparative HPLC (15 → 20% ACN/water gradient over 8 min), giving 4.8 mg (25%) of N-((3-chloro-8-methoxy-11H-indolo [3,2-c]quinolin-9-yl)methyl)-N-ethylethanamine (**9a**). HPLC rt = 2.19 min; *m/z* 368.2 [M+H]^+^.

**9b** (20% yield), **9c** (18% yield), and **9d** (20% yield) were prepared by the same method, which is as follows:

3-Chloro-8-methoxy-9-(piperidin-1-ylmethyl)-11H-indolo [3,2-c]quinoline (**9b**): HPLC rt = 2.40 min; *m/z* 380.3 [M+H]^+^.

3-Chloro-8-methoxy-9-((4-methylpiperazin-1-yl)methyl)-11H-indolo [3,2-c]quinoline (**9c**): HPLC rt = 2.58 min; *m/z* 395.3 [M+H]^+^.

N-benzyl-1-(3-chloro-8-methoxy-11H-indolo [3,2-c]quinolin-9-yl)methanamine (**9d**): HPLC rt = 3.16 min; *m/z* 402.1 [M+H]^+^.

### 2.3. FRET Melt/FRET Melt^2^

The NCI Diversity Set was screened by the traditional FRET Melt assay [39]. Briefly, a dual-labeled DNA oligomer probe (200 nM), bearing the G4-forming region of the specified promoter regions, was used to screen compounds of interest (10 μM). DNA was diluted in 10 mM Tris-HCl + 90 mM LiCl, 10 mM sodium cacodylate and 10 mM KCl, heated at 95 °C and rapidly cooled on ice. Annealed G4 DNA was added to a 96-well PCR plate, with or without test compounds, and incubated at room temperature for 30 min. Fluorescence was recorded from 25–95 °C, at every degree after a 30 s hold, on a Bio-Rad CFX Connect real-time PCR machine (Hercules, CA, USA). Examination of the novel indoloquinolines was performed with the enhanced FRET Melt^2^ assay [40], which involved an increase in labeled DNA to 2 μM and two notable additions to the described assay—(1) the addition of 10% glycerol to the solution and (2) one annealing cycle performed after the 30 min incubation period, but before recording fluorescence on the PCR machine in identical conditions to the above description and without reading fluorescence with a second annealing cycle as described, during which fluorescence was recorded. The importance of these two distinguishing processes is described in [40], wherein the Z′-score increased to >0.5. Due to a change in universities, the FRET Melt^2^ screening utilized a BioRad CFX96 thermocycler.

### 2.4. Electronic Crcular Dichroism (ECD)

The G4 oligonucleotides (5 µM) were annealed in 50 mM Tris-HCl (pH 7.4) with 10 mM KCl, without and with test compounds, by heating to 95 °C for 5 min and rapidly cooling on ice for 5 min. Spectra were collected with an Olis DSM-20 spectropolarimeter, equipped with a CD 250 Peltier cell holder (Bogart, GA, USA). Recordings were taken over the wavelength range 225–350 nm and at increasing temperatures (20–100 °C, every 10 °C, with a 1 min hold at temperature before spectra were recorded) in 1 mm quartz cuvette. The ordinate was reported as molar ellipticity (millidegrees); T_M_s were determined by performing a singular value decomposition (SVD) analysis, available with the Olis GlobalWorks software (Bogart, GA, USA), followed by non-linear regression fitting to determine the T_M_ using GraphPad Prism software (La Jolla, CA, USA).

### 2.5. Luciferase Assay

Human embryonic kidney cells (HEK-293), purchased from ATCC (Manassas, VA, USA), were cultured in 37 °C and 5% CO_2_ in Eagle’s Minimum Essential Medium (EMEM), enriched with 10% fetal bovine serum (FBS, Sigma Aldrich, St. Louis, MO, USA) and 1× penicillin/streptomycin solution (Gibco, Waltham, MA, USA). The cells were seeded in 24-well cell treated plates at 8x10^4^ cells/well and allowed to adhere overnight. While in exponential growth, the cells were transfected using Fugene HD transfection reagent (Promega, Madison, WI, USA) at a 3:1 ratio with the Renilla plasmid (pRL, 100 ng/well), co-transfected with 250 ng/well of either a luciferase vector containing no promoter (empty vector, EV) or one containing the full length KRAS promoter (FL [38]). The cells were also treated with up to 10 μM of the indicated compounds for 48 h, after which time cells were lysed using 1× Passive Lysis Buffer (Promega, Madison, WI, USA) and frozen at −20 °C. Following two freeze and thaw cycles to improve cell lysis, the luciferase expression was measured with the Dual Luciferase Assay kit (Promega) using a Lumat LB9507 luminometer. All experiments were performed in triplicate.

### 2.6. Cell Culture, Cytotoxicity and qPCR

The dose-dependent effect on cell viability was examined for AsPc1, Panc1, BxPc3 and MiaPaCa2 cells, all purchased by ATTC (Manassas, VA, USA). All cells were cultured in 37 °C and 5%CO_2_. BxPc3 and AsPc1 were grown in RPMI 1640 Medium (Roswell Park Memorial Institute 1640 Medium, ATCC, Manassas, VA, USA), while Panc1 and MiaPaCa2 cells were maintained in Dulbecco’s Modified Eagle Medium (DMEM, Lonza, Basel, Switzerland). All media were enriched with 10% fetal bovine serum (FBS, Sigma Aldrich) and 1× penicillin/streptomycin solution (Gibco). For cytotoxicity experiments, cells were seeded in 90 μL of media in a 96-well tissue culture-treated plates. AsPc1 cells were seeded at 250,000, 125,000 and 25,000 cells/mL for 24, 72 and 144 h time points, respectively, Panc1 cells were seeded at 100,000 and 20,000 cells/mL for 72 and 144 h experiments, respectively; BxPc3 were plated at 30,000 cells/mL for 144 h treatments and MiaPaCa2 cells were seeded at 40,000 cells/mL for 72 h experiments. Cells were allowed to adhere overnight before treatment with 10 μL of a 10× prepared serial dilution of compounds, as indicated. Compounds were diluted 1:3 over a 5-6 log-range, and cells were exposed to the compounds for the indicated times. For cells that were co-treated with oligonucleotides, at the time of compound treatment, the cells were also transfected with the indicated concentrations of PPRH ODN, encapsulated into liposomes with DOTAP (Sigma Aldrich St. Louis, MO, USA) at a 100:1 ratio. Changes in cell viability were measured with the Cell Titer AQeuous MTS assay (Promega; Madison, WI, USA), as directed by the manufacturer. After 2–4 h of incubation with the (3-(4,5-dimethylthiazol-2-yl)-5-(3-carboxymethoxyphenyl)-2-(4-sulfophenyl)-2H-tetrazolium salt) activated with phenazine methosulfate (PMS, Sigma Aldrich), absorbance was measured at 490 nm with a SpectraMax i3× Microplate reader (Molecular Devices; San Jose, CA, USA). Background absorbance was subtracted and absorbances were normalized to the untreated controls, before using GraphPad Prism to determine the IC_50_ for each drug/compound using non-linear regression models. All experiments were performed in biological triplicate with internal technical triplicates.

In order to examine the transcriptional effects, cells were plated at the same seeding density as described above, in 6- or 12-well tissue cultured plates. The next day, we treated the cells with the selected compounds, DMSO vehicle control, or just media. Next, 72 h later, the cells were harvested and the RNA was isolated with the GeneJet RNA purification kit (Thermo Fisher, Waltham, MA, USA). For the cDNA synthesis, 500 ng of RNA samples were reverse transcribed using the qScript cDNA reagent (Quantabio, Beverly, MA, USA). Quantitative real-time PCR was multiplexed on a Bio-Rad CFX96 thermocycler (Hercules, CA, USA) using TaqMan primers (Applied Biosystems, Carlsbad, CA, USA) for *KRAS* (FAM-labeled) and *GAPDH* as a housekeeping gene (VIC-labeled).

### 2.7. Three-Dimensional Cell Culture, Cytotoxicity, and Combination Chemotherapy Cytotoxicity

Cells were plated in ultra-low attachment plates (Corning Inc., Corning, NY, USA) in Dulbecco’s Modified Eagle Medium/Nutrient Mixture F-12 (DMEM/F12, ATCC, Manassas, VA, USA), supplemented with 1× B27 (Gibco, Waltham, MA, USA) 0.02 μg/mL epidermal growth factor (EGF, EMD Millipore, Burlington, MA, USA), solubilized in PBS and 0.02 μg/mL basic fibroblast growth factor (bFGF, Sigma Aldrich), solubilized in PBS. Three-dimensional growth was, thus, fostered for 5 days before cells were collected, and replated in an ultra-low attachment 96 well plate one night prior to compound treatment for 72 h, as described above; cytotoxicity was also determined using the MTS assay. The seeding density in 96 well plates for AsPc1, Panc1, BxPc3 and MiaPaCa2 cells was 500,000, 500,000, 90,000 and 40,000 cells/mL, respectively, as determined by cell seeding density experiments to ensure exponential cell growth (data not shown).

### 2.8. Statistics

GraphPad Prism was utilized for non-linear regression analyses and statistical comparisons. Dose-dependent changes in cell viability were compared to each other (e.g., 2D vs. 3D), using the extra sum-of-squares F test (*p* value set to 0.05) by comparing the IC_50_ values. Transcriptional effects and dose-dependent changes in luciferase were examined within each luciferase construct and were compared using a one-way ANOVA test with post-hoc Tukey analyses, as indicated in the text.

## 3. Results

### 3.1. NSC 317605 as a G4 Stabilizer

#### 3.1.1. Screening the NCI Diversity Set III

The NCI Diversity Set III of 1600 compounds was screened by the FRET Melt assay, using FAM-TAMRA dual-labeled KRAS G4_mid_ DNA. Of those 1600 compounds, 4 were identified with drug-like properties and reproducible shifts in the thermal stability of KRAS G4_mid_, which were as follows: NSC 317605 (an indoloquinoline), 274905 (a naphthalene pyridimidinamine), 106506 (a carbazole aceteamide) and 44750 (a dioxopyrrol benzoic acid). Further examination of these compounds for G4_mid_ stabilization via electronic circular dichroism (ECD) at a 2:1 ligand:DNA ratio, in the presence of 25 mM KCl, highlighted NSC 317605 as the most promising ligand with a ΔT_M_ of 18 °C, as compared to 5, 0, and 4 °C, respectively. Notably, NSC 317605’s indoloquinoline pharmacophore harbored similar traits to previously identified G4-stabilizing ellipticine and quindoline compounds, further highlighting its potential as a G4-interactive agent.

#### 3.1.2. Identification of Pan-KRAS G4-Stabilizing Compounds, and those Selective for G4_near_ or G4_mid_

NSC 317605 was further examined for its G4-interactive promiscuity in comparison to the known pan-G4 cationic porphyrin, TMPyP4, its isomeric negative control non-G4-interactive TMPyP2, the promiscuous NSC 176327, and the MYC promoter G4 interactive NSC 338258 and quindoline i. Each of these compounds was screened using the FRET Melt^2^ assay (with improved predictability [40]) against the MYC, KRAS G4_near_ and KRAS G4_mid_ dual-labeled FAM-TAMRA sequences (Figure 1A,B). TMPyP2, an isomeric negative control, was ineffective at stabilizing any of these G4s, whereas TMPyP4, a pan-G4-stabilizing agent, significantly increased all of their thermal stability. quindoline i [41,42] stabilized all three G4s, but was least “effective” with KRAS-G4_mid_. In agreement with the previous literature [43], NSC 338258 stabilized only the MYC G4. NSC 176327 stabilized both the MYC and the KRAS-G4_near_, but not the KRAS-G4_mid_, structures, whereas NSC 317605 stabilized KRAS-G4_mid_ and MYC G4, but not KRAS-G4_near_ (Table 2). Notably, both quindoline i and NSC 176327 have previously been identified as stabilizers of the MYC G4, but they both failed to have confirmed cellular activity related to the intracellular structure [41,43]. Due to the novelty of 317605 as an indoloquinoline, and its selectivity for the KRAS G4_mid_ structure, it was selected for further examination. TMPyP4 (25 μM) and 317605 (1 μM) were examined in a luciferase system that contained either no promoter (EV) or the full length KRAS promoter (KRAS/FL [38]), as well as in pancreatic cancer cell lines that harbored mutant KRAS^G12D^ proteins—Panc1 and AsPc1 (Figure 1C). Both the pan-G4-stabilizing TMPyP4 and KRAS G4_mid_-stabilizing NSC 317605 significantly decreased the promoter activity in the KRAS (by 65%), but not the EV plasmid. NSC 317605 effects on cellular viability at 72 h was measured by MTS in both cell lines (55 ± 3 and 26 ± 1 μM, in Panc1 and AsPc1 cells, respectively), which were then treated with the IC_50_s and effects on *KRAS* mRNA expression was measured. Similar to the luciferase findings, NSC 317605 significantly lowered *KRAS* expression in both cell lines. Changes in *MYC* expression were also monitored; at the cytotoxic IC_50_ doses in these cell lines, no change in *MYC* transcription was noted (data not shown).

### 3.2. Synthesis and Characterization of 317605 and Novel Indoloquinolines

#### 3.2.1. Synthesis of Indoloquinolines

NSC 317605 was (re)synthesized in order to confirm its potency and selectivity. It is well known that the purity and integrity of compounds in HTS screening sets can be less than optimal, sometimes leading to wasted efforts and inaccurate structure-activity assumptions. While this compound is known in the literature, the reported synthesis [44] involves Fisher indole synthesis, using a hydrazine that is not readily available and is difficult to prepare. With this in mind, we devised a convergent 7-step synthesis (Figure 1) that begins with the readily available 2-(hydroxymethyl)-4-nitrophenol (**1**) and 7-chloroquinolin-4-ol (**4**). Importantly, and in contrast to the original reported synthesis, this approach allows diversification of the amine tail in the final step, thereby allowing the rapid generation of analogs for structure-activity studies.

The synthesis of NSC 317605 began with iodination and chlorination of quinoline (**4**) resulted in a key intermediate **6**. Aniline **3** was readily prepared in high yields by methylation of **1,** followed by iron-promoted reduction. Treatment of **6** with the nucleophilic aniline in the presence of pyridine hydrochloride resulted in an S_N_-Ar reaction, giving compound **7**. A subsequent Pd-promoted intra-molecular Heck reaction resulted in the formation of two isomers of the requisite indoloquinoline in a ~10:1 ratio. The structure of the major isomer was confirmed to be 8 by 2D-NMR (Appendix A). After purification of 8 by preparative LCMS, the benzylic alcohol was oxidized with PDC and the resulting aldehyde underwent a reductive amination with diethyl amine to give the final product, **9a**. We confirmed that **9a** and NSC 317605 were identical by LCMS retention time and mass spectroscopy (Appendix A). Having confirmed the structure, we proceeded to prepare three additional analogs (**9b**–**9d**) using the same methodology. All final compounds were purified by preparative HPLC (can/water, C18 column) and the purity and structure were verified by LCMS (Figure 2 and Appendix A). We anticipate that this synthetic method will be highly versatile for the preparation of additional analogs in the future with diverse analogs of quinoline **4** and aniline **3**.

#### 3.2.2. Characterization of Indoloquinolines

The newly synthesized compounds, in addition to the NCI-obtained NSC 317605 (NSC), were screened for MYC and KRAS G4 stabilization using the optimized FRET Melt^2^ assay [40] (Figure 3A). Compounds NSC, **9a** and **9b** stabilized the KRAS G4_mid_ structure by +8, 6 and 7 °C, respectively, whereas compounds **9c** and **9d**’s ΔT_M_ for the KRAS G4_mid_ structure were <1 °C. None of the compounds demonstrated marked (>5 °C ΔT_M_) stabilization of the MYC G4. Importantly, the cutoff of 5 °C ΔT_M_ had been previously described from this assay in correlation with cellular activity and a mechanistic tie to G4 stabilization [40,41,43].

The new compounds were examined for their ability to modulate *KRAS* promoter activity in HEK-293 cells, stably transfected with either the promoterless vector (EV) or the *KRAS* full promoter vector (FL), following 48 h incubation (Figure 3B). None of the compounds demonstrated significant changes in promoter activity in the EV (range of normalized relative luciferase units—RLU—0.8 ± 0.2 to 1.2 ± 0.7 for 0.1 μM and 0.7 ± 0.3 to 1.1 ± 0.2 for 1 μM). Only compounds NSC and **9a**, namely the two identical compounds, significantly (*p* < 0.05) decreased KRAS promoter activity at 1 μM by 34 ± 11 and 46 ± 23%, respectively. Changes in RLU at 0.1 μM ranged from −7 ± 25 to +6 ± 30%; at 1 μM, compounds **9b**–**9d** insignificantly decreased normalized RLU by 8 ± 25 to 30 ± 16%, respectively.

Compounds that stabilized the KRAS G4 in the FRET Melt^2^ assay were further tested in KRAS-dependent AsPc1 pancreatic cancer cells for their 72 h cytotoxicity (Figure 3C) and their correlating regulation of *KRAS* transcription (Figure 3D). In accordance with the luciferase findings, compounds NSC and **9a** significantly decreased *KRAS* expression at their cytotoxic doses to 0.45 ± 0.23 and 0.32 ± 0.02-fold, as compared to the DMSO control, respectively. Compound **9b**’s vehicle normalized expression of 0.88 ± 0.29 was not significantly different than DMSO or the untreated (1.02 ± 0.31) controls. Changes in transcription tied to cytotoxicity were also examined in Panc1 pancreatic cancer cells, wherein trends indicating a decrease in *KRAS* with compounds NSC and **9a** were also observed (vehicle normalized expression was noted to be 0.57 ± 0.36 and 0.55 ± 0.34), but significance was not noted. Compound **9b**’s normalized expression of 1.36 ± 0.94 was not notably different than either vehicle or untreated (0.99 ± 0.24) controls, in agreement with the luciferase and AsPc1 findings.

### 3.3. Activity of Compound ***9a*** in Pancreatic Cancer Cell Lines

#### 3.3.1. Compound **9a** Cytotoxicity Is Enhanced by G4_mid_-Stabilizing PPRH Oligonucleotides in KRAS-Dependent Pancreatic Cell Lines

The time-dependent cytotoxicity of the lead compound, **9a**, was examined in AsPc1 cells, which harbor the G12D KRAS mutation and demonstrated high addiction to KRAS signaling. There was a time-dependent decrease in the IC_50_ of compound **9a**, from 133 ± 39 to 54 ± 20 to 22 ± 0.5 μM at 24, 72, and 144 h, respectively (Figure 4A). A direct comparison of 72 h IC_50_s for NSC 317605 and compound **9a** demonstrated no significant difference in response (Appendix A). Cytotoxicity in a broader pancreatic panel, including Panc1 cells with the G12D KRAS mutation and moderate addiction to KRAS signaling and BxPc3 cells with wild-type KRAS and no apparent addiction to KRAS signaling, was further examined at 144 h (Figure 4B). Despite the variations in KRAS mutation status and sensitivity, all of the cell lines showed comparable sensitivity, with IC_50_s calculated to be 14 ± 3 and 27 ± 1 μM in Panc1 and BxPC3 cells, respectively. While in the current study, selectivity for KRAS G4_mid_ over G4_near_ and MYC was observed, only a small subset of target G4s were examined and the cytotoxicity across pancreatic cancer cell lines is most likely related to promiscuity of G4 binding and stabilization for compound **9a**, as frequently noted with G4-interactive compounds.

We have recently described a series of polypurine reverse Hoogsteen (PPRH) oligonucleotides that interact with C-rich DNA that complements G–rich regions of the KRAS promoter, and synergistic activity of G4_mid_ that targets PPRH1 and PPRH2 with NSC 317605 in AsPc1 cell lines [45]. We confirmed the synergy between these PPRHs and compound **9a**, and expanded the study to the expanded pancreatic cancer panel (Figure 4C). Dose–dependent lowering of compound **9a** cytoxicity, as measured by fold changes in the 144 h IC_50_, was noted in the two cell lines that harbor mutant KRAS and a measure of addiction to aberrant signaling—AsPc1 and Panc1—of up to 92 and 75%, respectively. No marked sensitization of cytoxocity was identified in the BxPc3 cell lines (maximum decrease in the IC_50_ was measured to be 22%), which is consistent with this cell line’s lack of dependency on KRAS and the presumed broader G4 binding of compound **9a**.

#### 3.3.2. Compound **9a** Demonstrates Enhanced Cytotoxicity in a Three-Dimensional Morphology of MiaPaCa2 Pancreatic Cancer Cells

To increase the clinical correlation of in vitro work, cytotoxicity of gemcitabine (the cornerstone chemotherapy for pancreatic cancer) was examined in the panel of pancreatic cancer cell lines in two– (traditional attached, as described above) and three-dimensional (using low attachment plates and media supplemented with B27, EGF and bFGF) morphologies. In the 3D model used, cells were allowed to form nascent structures with associated necrosis and hypoxia in ultra-low attachment plates, rather than facilitating perfect spheroid formation, with either droplet formation or Matrigel/agarose [46,47,48]. MiaPaCa2 cells were added to this panel of cell lines. These cells harbor the most frequent KRAS mutation that occurs in pancreatic cancer (G12D) and demonstrate moderate addiction to KRAS signaling that is comparable to the Panc1 cells [49,50,51]. All cells were exposed to a range of gemcitabine in 2D and 3D morphologies for 72 h; overall, the 2D cells responded in a cytostatic manner with lower plateaus in their dose response, ranging from 44–75% of control cell growth (Figure 5A). Accounting for the cytostatic response, IC_50_s for 2D conditions were 0.2, 0.15, 0.02 and 0.04 μM for AsPc1, Panc1, MiaPaCa2 and BxPc3 cells, respectively. AsPc1, Panc1 and BxPc3 cells grown in 3D-promoting conditions did not respond at all to gemcitabine in a manner significantly (*p* < 0.05, as determined by comparison of best fit of non-linear regressions) different than their matched 2D conditions. Intriguingly, 3D–grown MiaPaCa2 cells not only responded to gemcitabine in a cytostatic manner, but also were significantly more responsive (*p* < 0.05, as determined by comparison of best fit of non–linear regressions) than their 2D grown counterparts, with a two-fold lower IC_50_ of 0.01 μM.

Following this interesting characteristic of the MiaPaCa2 cells, cytotoxicity (72 h) of an array of chemotherapeutic agents utilized in the clinical management of pancreatic cancer was examined, including the active metabolite of irinotecan, SN38, 5–fluorouracil, oxaliplatin, gemcitabine and paclitaxel (Figure 5B, Table 3). The IC_50_ of oxaliplatin was significantly (2.2–fold, *p* < 0.01) higher in 3D–, as compared to 2D–, grown cells; however, no marked difference in drug efficacy was noted with any other drug examined in MiaPaCa2 cells grown in the 3D, as compared to 2D, morphology. Compound **9a** was examined under these same 2D and 3D conditions for 72 h. As compared to the AsPc1 cell line above, attached MiaPaCa2 cells were markedly less responsive, with an IC_50_ of 163 μM. Remarkably, 3D–grown MiaPaCa2 cells were significantly (3.8–fold, *p* < 0.01, as determined by comparison of best fit of non-linear regressions) more responsive to compound **9a**. Combinatorial studies were completed with compound **9a** and gemcitabine, but no synergy was noted (data not shown). To our knowledge, this is the first-time enhanced sensitivity to a drug or compound has been noted in 3D–grown pancreatic cancer cells.

## 4. Discussion

The current study screened the NCI DTP Diversity Set III of 1600 compounds for *KRAS* G4_mid_ stabilization; four compounds with drug-like properties and reproducible shifts in thermal melt profiles were identified. Of these compounds, NSC 317605 produced the most marked thermal stabilization of the mid-G4 and fit into the privileged chemical space of several G4-interactive compounds [41,42,43,52,53]. Moreover, NSC 317605 cellular cytotoxicity correlated with inhibition of transcription in two pancreatic cancer cell lines. A small expansion of the indoloquinoline pharmacophore of NSC 317605 was explored with various amines, but the parent diethylamine compound (synthesized as **9a**) was demonstrated to have the lead selectivity and cellular activity. Examination of **9a**’s efficacy in a panel of pancreatic cancer cell lines with varied mutational status and reliance on KRAS suggests compound binding to other G4 structures, as there was no identified correlation with KRAS dependency and compound **9a**’s activity. There was, however, synergy noted with compound **9a** and selective KRAS G4_mid_-stabilizing PPRHs, and subsequent transcriptional downregulating [45]; this synergy was only evident in KRAS-dependent cell lines. Studies were further performed in an expanded pancreatic cancer cell line panel in two- and three-dimensional growth conditions, wherein differential sensitivity to gemcitabine was identified between the growth conditions in AsPc1, Panc1 and BxPc3 cells. Interestingly, MiaPaCa2 cells also showed differential sensitivity to gemcitabine between the two morphological growth conditions, but the 3D cells were more sensitive, rather than the decreased sensitivity noted in the other cell lines. This phenomenon of enhanced sensitivity in 3D conditions was not ubiquitous across chemotherapies, with no varied effects for paclitaxel, SN38 or 5FU and decreased sensitivity to oxaliplatin. Of the panel of cell lines studied, MiaPaCa2 were the least responsive to compound **9a** in 2D conditions; however, a remarkable 4-fold sensitization was noted in the 3D growth conditions. To our knowledge, this is the first identification of such a marked enhancement of compound efficacy in 3D, as compared to 2D growth conditions. Furthermore, while the downregulation of KRAS has been shown to modulate spheroid formation [54], this demonstration of enhanced efficacy of KRAS modulation in 3D morphology is also novel and highlights the importance of studying cancers in an array of morphologies and particularly, in those closer to physiological relevance for patients.

The indoloquinoline scaffold falls in the privileged chemical space that is home to several reported G4 stabilizers [53]. Solution-phase NMR structural data for one of these G4 stabilizers, indolo [2,3-b]quinoline, demonstrate that this series of compounds does not directly intercalate into the G4 structure, but rather pi-pi stacks on both the upper and lower face of the G4 structure and recruits a 3′ and/or 5′ flanking base in order to form a transient binding pocket [55]. The recruitment of the 3′ and/or 5′ flanking bases expands the opportunity for sequence specificity imparted by loop and flanking nucleobases. The structural information for indolo [2,3-b]quinoline strongly suggests that there is the opportunity to gain sequence selectivity both by the specific shape and properties of the core *and* by the secondary interactions facilitated by solubilizing “arms” that reach out from the core. In the current work, we describe an efficient synthetic route (Figure 1) to NSC 317605 and the related analogs. While the parent compound remains the lead, future studies that expand this rich chemical set of G4 stabilizing compounds are likely to lead to novel and more potent compounds.

Polypurine reverse-Hoogsteen (PPRH) ODNs are DNA molecules composed of two symmetrical stretches of polypurines separated and connected by five thymines, forming a hairpin structure, wherein Hoogsteen base-pairing facilitates G:G and A:A bonds. These structures further utilize Watson–Crick base-pairing to form triplex structures with cytosine-rich regions of DNA, freeing the guanine-rich complementary strand to facilitate higher order formations, such as G4s [56,57,58,59,60,61,62]. We previously detailed a series of PPRH ODNs binding to G-rich regions of the KRAS gene, particularly within the promoter region, that mediated KRAS downregulation and subsequent cytotoxicity [38]. We also described the synergy between NSC 317605 and PPRHs that enhances the formation of G4_mid_. In the current work, the synergy of G4-enhancing PPRH’s and compound **9a** was confirmed in the AsPc1 cell line and demonstrated in the moderately KRAS-dependent Panc1 cells, but not in the KRAS-independent BxPc3 cells [49,50,51]. This correlation with KRAS-dependency is notable, as compound **9a** appears to bind and stabilize more than just the KRAS G4_mid_ structure, as evidenced by its equipotency in KRAS-dependent and -independent cell lines, but its selectivity can be enhanced with the combination of G4-stabilizing PPRH ODNs [45]. This combinatorial approach to G4-stabilization is novel and can be applied on a broad scale, with G4-stabilizing compounds of various degrees of promiscuity, wherein selective gene selectivity can be enhanced for greater efficacy and tolerability.

Within the KRAS promoter, several G-rich regions exist. Stable and inducible G4s have been demonstrated within two distinct regions—G4_near_ and G4_mid_. The near G4-forming region was previously examined and various G4 formations were proposed to exist in equilibrium [34,35,36,37,63], which was clarified and the predominant structure was shown to be a unique parallel G4, with a kinked out thymine in one continuous guanine run. The G4_mid_ structure was shown to be a mixed parallel/antiparallel structure with longer loops. Our previous study indicated that G4-mediated transcriptional silencing within the KRAS promoter can be mediated by G4_mid_ [38], which was confirmed by the current work. A compound was identified that selectively stabilized the KRAS promoter mid G4, which subsequently led to cellular activity and silencing of KRAS transcription. While compound NSC 176327 did selectively stabilize G4_near_ over G4_mid_, it has been previously shown to not have correlating cellular activity related to any G4 [41,43], and was not felt to be strong enough of a candidate to dissect the physiological function of G4_near_. Thus, while the data presented continue to confirm the biological activity of G4mid, our work does not negate the previous findings related to G4_near_. Rather, we describe a new molecular target for future drug discovery efforts, with a strong likelihood of correlating with clinical outcomes.

KRAS is a well-validated therapeutic target in oncology, and one that is no longer considered “undruggable”, with the new clinical agents that target the G12C mutant protein [64]. While the development of sotorasib and adagrasib are remarkable, G12C is common only in non-small cell lung cancer, and these drugs are not utilized in pancreatic or colorectal cancers often [65,66]. Moreover, neither compound has benefits for patients with KRAS amplification or overexpression, so there remains a need for the development of novel compounds that can modulate KRAS. It has been demonstrated that downregulation of all KRAS isoforms has lethality to cancer cells with aberrant KRAS signaling, and that there is a large potential therapeutic window for other cell types [1,18,19,20,21]. Thus, targeting the regulation of transcription through stabilized G4 structures by small molecules and targeted PPRH oligonucleotides that selectively inhibit transcription of *KRAS* will have great potential in ultimately achieving clinical activity in patients whose tumors have dysregulated KRAS function, such as addicted cancers and RASopathies.

## Data Availability

Not applicable.

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
