# Peer review of "Indoloquinoline-Mediated Targeted Downregulation of KRAS through Selective Stabilization of the Mid-Promoter G-Quadruplex Structure"

_genes, 2022, doi:10.3390/genes13081440_

Round 1

Reviewer 1 Report

In their manuscript, Psaras et al. have used a FRET Melt Assay to screen the NCI Diversity Set III library of 1600 compounds for their ability to stabilise the G-quadruplex of KRAS G4mid. They found that NSC 317605, an indoloquinoline, stabilised KRAS G4mid but not KRAS G4near. They synthesised NSC 317605, called 9a, and made three analogues of 9b-d. The authors investigated the bioactivity of the synthesized compounds in pancreatic cancer cells and decided to focus on 9a.
This study reports some interesting data on G4 compounds that distinguish between G4 conformations in the KRAS promoter. The manuscript shows some important methodological limitations that need to be addressed before it can be considered for publication.
1) Fig. 1B: Please provide the values of TM in a table, as they are not readily obtainable from the graphs. The idea of performing the experiments in Li+, which is not compatible with G4, is questionable. Are the authors sure that the sequences in Li+ fold into a G4 in the presence of the compounds?  Imino proton analysis would help to this regard

2) Fig. 1 C ( top ). Using an empty vector (luciferase driven by NO promoter) is not good practise. It is obvious that the compounds have no effect on an empty vector (How much luciferase is expressed by a vector without a promoter? Simply a background). In addition, the luciferase expressed by the transfected cells not treated with the compounds is missing (reference is missing). A better way to design the luciferase experiment is to introduce point mutations in G-near and G-mid and then measure their effects on luciferase. In this case, the reference would be luciferase driven by the full KRAS promoter and not luciferase from an empty vector without  promoter. Most importantly, the authors should perform dose-response experiments, using increasing amounts of the designed compounds, to investigate the effects on luciferase of the compounds. The graph 1C top reporting luciferase at only one compound concentration (which one?) is not enough for drawing any conclusion.

3) Fig. C middle. Literature reports that suppression of KRAS gene reduces cell viability. Why does compound 317605, which downregulates G4-mid and not G-near, have an IC50 of about 12 micromol/L, whereas compound 176327, which downregulates G-near and not G4-mid, has an IC50= of 0.4 micromol/L? I would have expected the opposite, i.e cells more sensitive to 176327 than 317605. The viability data are not in keeping with the luciferase data. This may depend on the questionable way the luciferase data have been obtained.

4) Fig.1C bottom. The error bars relative to 176327 are too large. The experiments should be redone. Also, a dose-response effect must to be reported. To draw any conclusion, the protein KRAS must be also measured by Western blot.

5) Fig.4AB. These figures are not clear. In colour, the figure should improve in clarity.   

5) Fig. 3C and Fig. 1C middle, cell viability data. Compound 9a is identical to compound NSC 317605. Fig. 1c middle shows that the IC50 of NSC 317605 in AsPc1 cells is 12 micromol/L, while Fig. 3C shows that the IC50 of 9a (the same compound) in AsPc1 cells is 60-70 micromol/L. There is no agreement between the data.

6) Page 12 manuscript: KRAS G12C is not the most frequent mutation in pancreatic cancer. The most frequent is KRAS G12D.

Final observation: The human KRAS has a rather complex promoter harbouring three G4 motifs that can fold into different G4s. From the literature, it appears that G4 near acts as a kind of hub for the recruitment of transcription factors. G4 mid probably has such a function, as it binds strongly to PARP1. Unfortunately, the structural polymorphism of the KRAS promoter upstream TSS is quite complex and difficult to study. The authors concluded from Fig 1C data, which are not strong and unclear, that G4 near has no effect on the KRAS promoter, whereas G4 mid does have, although several studies report strong binding of transcription factors to G4-near.
In conclusion, the manuscript reports an indoloquinoline that discriminates between G4 conformations: an interesting finding. But, the part of the manuscript regarding the luciferase data of in Fig. 1C top is confusing and either new experiments are done as suggested by this reviewer or the data should be removed. A part of the discussion should be revised, as well. With this revision, the manuscript should improve in terms of linearity and clarity.

Author Response

We wish to thank the reviewer for their comments and suggestions.  We are glad that the data were interesting and novel. Before delving into specific responses, it is notable that we are unable to perform any additional studies with NSC 176327 due to a lack of compound availability from the NCI. Thus, westerns are unfortunately not possible. The PI, however, was able to take the opportunity to re-analyze the raw data previously obtained and updated the text as described below. While we were limited in additional experiments with 176327, we believe that the additional analyses and figures as well as clarified text help address the concerns. Overall, in an effort to improve the manuscript for publications and in light of the reviewer’s suggestions, we have made the following edits:

Comments and Suggestions for Authors

Fig. 1B: Please provide the values of TM in a table, as they are not readily obtainable from the graphs. The idea of performing the experiments in Li+, which is not compatible with G4, is questionable. Are the authors sure that the sequences in Li+ fold into a G4 in the presence of the compounds?  Imino proton analysis would help to this regard

Response: Tm values have been added in Table 2. Li+ is standard for Fret Melt in all of its iterations, and is supplemented by both Na+ and K+. See PMID 17472900, 26079222, 33368198 and 31891781.

Fig. 1 C ( top ). Using an empty vector (luciferase driven by NO promoter) is not good practise. It is obvious that the compounds have no effect on an empty vector (How much luciferase is expressed by a vector without a promoter? Simply a background). In addition, the luciferase expressed by the transfected cells not treated with the compounds is missing (reference is missing). A better way to design the luciferase experiment is to introduce point mutations in G-near and G-mid and then measure their effects on luciferase. In this case, the reference would be luciferase driven by the full KRAS promoter and not luciferase from an empty vector without promoter. Most importantly, the authors should perform dose-response experiments, using increasing amounts of the designed compounds, to investigate the effects on luciferase of the compounds. The graph 1C top reporting luciferase at only one compound concentration (which one?) is not enough for drawing any conclusion.

Response: Compounds can have an effect on output from the luciferase assay from either an EV or an SV40 driven promoter. Notably the porphyrin compounds TMPyP2 and TMPyP4 have a demonstrated “dampening” of the luciferase glow due to the absorbance of the compound (see PMID 26597160). There is literature to support inclusion of effects in an EV as a control, albeit expected to be for background control. Point mutations within the G4 near and G4 mid sequences were previously reported, and effects from TMPyP4 were measured (same PMID).  As mentioned above, NSC 176327 is no longer available, so we are unable to add studies in these in the point mutation plasmids, or to perform a dose response.  Specific doses used in the study were added to the text in lines 318-319.

Fig. C middle. Literature reports that suppression of KRAS gene reduces cell viability. Why does compound 317605, which downregulates G4-mid and not G-near, have an IC50 of about 12 micromol/L, whereas compound 176327, which downregulates G-near and not G4-mid, has an IC50= of 0.4 micromol/L? I would have expected the opposite, i.e cells more sensitive to 176327 than 317605. The viability data are not in keeping with the luciferase data.

Response: Thank you for asking this clarifying question, as it is a major point. Stabilization of G4mid and not G4near was previously shown to contribute to decreased promoter activity when assays were completed with the entire core promoter (see PMID 26597160 as compared to assays done with minimal portions of the promoter with studies of G4near).  In the current study, we switched from the genetic alterations to selective compounds and identified the two aforementioned compounds that differentiate near vs mid and have effects on KRAS regulation in agreement with the previously published genetic experiments.  These compounds, however, are not selective for only KRAS G4s. NSC 176327 has been shown to stabilize the MYC promoter G4 (PMID 2195615), and intercalates DNA in ds form lending to greater overall cytotoxicity unrelated to the KRAS G4.

Fig.1C bottom. The error bars relative to 176327 are too large. The experiments should be redone. Also, a dose-response effect must to be reported. To draw any conclusion, the protein KRAS must be also measured by Western blot. ... Fig. 3C and Fig. 1C middle, cell viability data. Compound 9a is identical to compound NSC 317605. Fig. 1c middle shows that the IC50 of NSC 317605 in AsPc1 cells is 12 micromol/L, while Fig. 3C shows that the IC50 of 9a (the same compound) in AsPc1 cells is 60-70 micromol/L.

Response: As mentioned, we are unable to perform any additional studies with 176327, including western blots. We were, however, able to take this opportunity for the PI to re-examine the raw data from these experiments and to address the concern by analyzing the data with all biological replicates included as individual points and any technical replicates as the summary data. In doing so, we further ensured that any and all statistical comparisons were done with GraphPad Prism using the same formula and parameters (previous iterations were performed on different versions of Prism and thus had varied analyses such as three or four parameters). In doing so, several points were revised – see an updated Figure 1, text 325-327, 442-444 and new Figure S3.

Fig.4AB. These figures are not clear. In colour, the figure should improve in clarity. And Page 12 manuscript: KRAS G12C is not the most frequent mutation in pancreatic cancer. The most frequent is KRAS G12D.

Response: both of these items were updated/addressed.

Reviewer 2 Report

The manuscript by Psaras et al. addresses the notable lack of specific G-quadruplex binding agents that limit medical applications strategically set on drug-targeting of specific structures adopted by guanine-rich segments in oncogene promoters.  By relying on 1600 compounds from NCI DTP Diversity Set III, four candidates marked by promising drug-like properties were identified and amongst NSC 317605 was selected for further studies, based on its outstanding G-quadruplex stabilization. This effect was evaluated by considering two KRAS- and one C-MYC-related G-quadruplexes-forming guanine-rich DNA models by correlating the biophysical results obtained for the (analogues of) NSC 317605 with the results of studying how the compounds affect promoter-dependent transcription and evaluation of the cytotoxic effects in different cell lines. The study of the compounds of interest were performed in parallel to control experiments, where previously characterized G-quadruplex-binding agents were used. The manuscript clearly reflects admirable level of systematic research approach. In particular, upon considering the ‘promiscuity of G-quadruple binding’ the authors employ previously demonstrated synergistic activities of polypurine reverse Hoogsteen oligonucleotides and G-quadruplex-binding agents to corroborate preferential binding of the lead molecule to G4mid versus other G-quadruplex forming segments. The interpretation of data is solid and followed by adequate discussion, putting the newly derived indoloquinoline analogues in a wider context, in particular by considering their G-quadruplex stabilizing effects and dose-dependent outcomes in biological assays. Commendably, the synthetic route is reported, applicable to design and broaden G-quadruplex binding compounds apart the 9a-d that are assessed in the manuscript. Somehow aside the main focus the manuscript reports peculiar finding on that the chemotherapy drug gemcitabine effects might be dependent on cell morphology, suggesting that evaluation of anti-cancer compounds could in general be more insightful upon considering different morphology growth conditions.

Comments:

At several points compounds ‘NSC-317605’ and ‘9a’ are noted as ‘the same’. Yet, their varying effects is evident, and although the authors give remark that this might be related to the different approaches by which the compounds were synthesized, no specific reason or explanation is given, for example presence/concentrations of different counterions, residual solvent/by products etc. The revised manuscript should include a brief comment to address the above concern.

Minor remarks:

1. rephrase Table 1 title to: Sequences of oligonucleotides used in the current study

2. Parameters-details (timing and temperature range) should be provided for the annealing cycle performed before recording fluorescence. If this annealing procedure (before FRET melt analysis) included slow cooling, while the sample preparation for the electronic circular dichroism experiments were prepared by heat-and-quench procedure, the authors should briefly address whether/how protocol differences might have been reflected in the results. 

3. Acronyms must be defined when first mentioned, including ‘ECD’ for Electronic circular dichroism.

4. Graphics should be improved, in particular:

- higher resolution needed in Figure 1

- avoid partial marking in vertical axis in Figure 3

                - in Figure 4 A symbols (full-colored triangles, circles and squares) need to be changed, as in the current presentation it is impossible to resolve data at certain points

5. The list of references needs to be revised in order to correct a number of inconsistencies, including authors’ names and journals’ abbreviations.

Author Response

We wish to thank the reviewer for their comments and suggestions.  We are glad that the data and manuscript were found to be clear, systematic, and solid. In an effort to improve the manuscript for publications and in light of the reviewers suggestions, we have made the following edits:

Comment: At several points compounds ‘NSC-317605’ and ‘9a’ are noted as ‘the same’. Yet, their varying effects is evident, and although the authors give remark that this might be related to the different approaches by which the compounds were synthesized, no specific reason or explanation is given, for example presence/concentrations of different counterions, residual solvent/by products etc. The revised manuscript should include a brief comment to address the above concern.

Response: While we noted the same anomaly, we had no previously reanalyzed the data in a side by side comparison. During this revision, we were able to reanalyze the data from NSC 317605, and to do a side-by-side comparison with compound 9a. Notably, the reanalysis of the NSC data with the same non-linear regression model as that done with compound 9a led to a better fit of the IC50 (as evidenced by a smaller 95% confidence interval) and the data was updated in the text on lines 323-325. Moreover, a direct comparison of cytotoxicity in AsPc1 cells was added (Figure S3) and referred to in the text on lines 440-442. Notably, there is no significant difference in cell efficacy between the compounds.

Minor remarks:

  1. rephrase Table 1 title to: Sequences of oligonucleotides used in the current study

 Response: Our plurality is fixed.

  1. Parameters-details (timing and temperature range) should be provided for the annealing cycle performed before recording fluorescence. If this annealing procedure (before FRET melt analysis) included slow cooling, while the sample preparation for the electronic circular dichroism experiments were prepared by heat-and-quench procedure, the authors should briefly address whether/how protocol differences might have been reflected in the results.

Response: we have added clarifying details in the text to clarify. All G4s were annealed with rapid cooling. The additional annealing process in FRET Melt is an entire melt and anneal cycle as performed in the FRET melt itself (same ramp rate, just minus fluorescence readings), following the described G4 annealing and done in the presence of the compounds. Text was added on lines 197-200 describing this, as well as explaining why it is done.

  1. Acronyms must be defined when first mentioned, including ‘ECD’ for Electronic circular dichroism.

Response: fixed

  1. Graphics should be improved, in particular: higher resolution needed in Figure 1, avoid partial marking in vertical axis in Figure 3, in Figure 4 A symbols (full-colored triangles, circles and squares) need to be changed, as in the current presentation it is impossible to resolve data at certain points

Response: all graphics were updated as suggested.

  1. The list of references needs to be revised in order to correct a number of inconsistencies, including authors’ names and journals’ abbreviations.

Response: EndNote was used to make the reference list using a protocol. We will work with the journal to ensure that we follow their reference guidelines appropriately.  Thank you.

Round 2

Reviewer 1 Report

 The authors have not addressed any of the critical comments I made in my previous review. I regret that the manuscript still has the limitation that the conclusions are not supported by the experimental data. Some of the data presented are weak and do not support the conclusions. As suggested in my previous review, the authors should simply report their findings without making a comparison between the KRAS G4s. Such a comparison is possible, but only on the basis of more solid data and better designed experiments, as I pointed out in my first review. The data on cell viability as a function of G4 ligand concentration clearly show that the G4near motif is much more sensitive than the G4mid motif, which is in clear contrast to the authors' conclusions. This should have alerted the authors that their conclusions based on a simple luciferase assay, designed in a non-convincing way,  could be unreliable. 

Author Response

We understand the reviewer's concerns and regret that we are unable to perform additional studies with NSC 176327 due to its lack of availability.  We have updated the manuscript to focus only on 317605 in cellular studies, siting the literature findings with 176327 (and quindoline i) as compounds that stabilize structures in cell free experiments, but that fail to have intracellular mechanisms related to G4-stabilization (manuscript lines 317-332 and updated figure 1). The discussion was also edited to address the conclusions that can be drawn from the data (lines 623-640).